# Improving Discriminative Multi-Modal Learning with Large-Scale Pre-Trained Models

## Abstract

This paper investigates how to better leverage large-scale pre-trained uni-modal models to further enhance discriminative multi-modal learning. Even when fine-tuned with only uni-modal data, these models can outperform previous multi-modal models in certain tasks. It's clear that their incorporation into multi-modal learning would significantly improve performance. However, multi-modal learning with these models still suffers from insufficient learning of uni-modal features, which weakens the resulting multi-modal model's generalization ability. While fine-tuning uni-modal models separately and then aggregating their predictions is straightforward, it doesn't allow for adequate adaptation between modalities, also leading to sub-optimal results. To this end, we introduce **M**ulti-**M**odal **Lo**w-**R**ank **A**daptation learning (**MMLoRA**). By freezing the weights of uni-modal fine-tuned models, adding extra trainable rank decomposition matrices to them, and subsequently performing multi-modal joint training, our method enhances adaptation between modalities and boosts overall performance. We demonstrate the effectiveness of MMLoRA on three dataset categories: audio-visual (e.g., AVE, Kinetics-Sound, CREMA-D), vision-language (e.g., MM-IMDB, UPMC Food101), and RGB-Optical Flow (UCF101).

## 1    Introduction

Large-scale pre-trained models have exhibited remarkable performance across various downstream tasks (Radford et al., 2021; Brown et al., 2020). This trend has also been validated across different modalities, including language (OpenAI, 2023; Anil et al., 2023; Touvron et al., 2023), vision (Radford et al., 2021; Oquab et al., 2023), and audio (Girdhar et al., 2023; Huang et al., 2022a). Building on this success, Multi-modal Large Language Models (MLLM) have emerged. By connecting pre-trained language and vision models, MLLMs equip language models with the ability to 'see' (Dai et al., 2023; Liu et al., 2023a). While their primary focus lies in text generation and dialogue, we delve into the utilization of large-scale pre-trained models in discriminative multi-modal learning.

Large-scale pre-trained models significantly enhance discriminative multi-modal learning performance (as shown in Sec 4.2.2). In fact, simply fine-tuning these models with corresponding uni-modal data often outperforms recent multi-modal models. As Table 1 illustrates, fine-tuning pre-trained vision and audio models enables uni-modal models to surpass the performance of previously proposed multi-modal models (Fan et al., 2023; Peng et al., 2022) on AVE, Kinetics-Sound, and CREMA-D. Similarly, for MM-IMDB, by fine-tuning an advanced language model, the uni-modal approach can surpass the performance of recent multi-modal models (Li et al., 2023b).

However, despite the power of large-scale pre-trained models, when applied to multi-modal joint training, they can lead to insufficient learning of uni-modal features. Specifically, during linear evaluation, encoders from multi-modal learning underperform compared to their uni-modal fine-tuned counterparts (as Table 3 shows). This issue is called Modality Laziness (Du et al., 2023) or Modality Competition (Huang et al., 2022b), which has been proven to affect the generalization performance of the resulting multi-modal models. Uni-Modal Ensemble (UME) provides a straightforward solution by simply aggregating predictions from separately learned uni-modal models. However, this approach lacks cross-modal interaction, which may result in inadequate learning of paired features (Du et al., 2023) and subsequently lead to sub-optimal performance.

Table 1: **Fine-tuning large-scale pre-trained models with uni-modal data.** We report the performance of fine-tuned uni-modal models and prior multi-modal models. The performance of the multi-modal models on AVE and CREMA-D is sourced from Fan et al. (2023) (CVPR'23), on MM-IMDB from Li et al. (2023b) (CVPR'23) and on Kinetics-Sound from Peng et al. (2022) (CVPR'22). The reported evaluation metrics are Top-1 Accuracy (AVE, Kinetics-Sound and CREMA-D) and F1-Micro/F1-Macro (MM-IMDB). We have **bolded** the performances of uni-modal models that outperform previous multi-modal models.

| Dataset Model | AVE | Kinetics-Sound | CREMA-D | MM-IMDB |
|---|---|---|---|---|
| Recent Multi-modal Model | 68.1 | 63.1 | 65.3 | 66.7 / 61.7 |
| Fine-tuned Visual Model | **88.1** | **84.3** | **77.7** | 60.2 / 52.6 |
| Fine-tuned Audio / Language Model | **85.6** | **69.6** | **75.8** | **68.6 / 63.9** |

Considering that uni-modal fine-tuned models already capture a significant amount of features, we hypothesize only a limited number of parameters are required for cross-modal adaptation. To this end and taking inspiration from Parameter-Efficient Fine-Tuning, particularly LoRA (Hu et al., 2021), we introduce a novel approach called **M**ulti-**M**odal **Lo**w-**R**ank **A**daptation learning (**MMLoRA**). This method begins by freezing the weights of the uni-modal fine-tuned models and then introduces additional trainable rank decomposition matrices to a specific modality or all modalities' models. Subsequently, it proceeds with multi-modal joint training. During joint training, these newly introduced parameters facilitate improved adaptation between modalities, resulting in collaborative enhancements in predictions. MMLoRA not only surpasses other methods but also outperforms its fully fine-tuned counterpart. We demonstrate the effectiveness of MMLoRA across three categories of multi-modal datasets: audio-visual datasets [AVE (Tian et al., 2018), Kinetics-Sound (Arandjelovic & Zisserman, 2017), and CREMA-D (Cao et al., 2014)], vision-language datasets [MM-IMDB (Arevalo et al., 2017) and UPMC Food101 (Wang et al., 2015)], and the RGB-Optical Flow action recognition dataset [UCF101 (Soomro et al., 2012)].

## 2 RELATED WORK

**Large-Scale Pre-Trained Models.** Models pre-trained on large-scale datasets have consistently demonstrated exceptional performance when fine-tuned for downstream tasks (Devlin et al., 2018; Yang et al., 2019; He et al., 2020). They have even showcased remarkable results in few-shot or zero-shot testing scenarios (OpenAI, 2023; Brown et al., 2020; Radford et al., 2021; Anil et al., 2023). Widely-used and effective pre-training methods include Generative Pre-Training (Radford et al., 2018; Chen et al., 2020a), Mask Data Modeling (He et al., 2022; Devlin et al., 2018), Contrastive Learning (Radford et al., 2021; Cherti et al., 2023; Girdhar et al., 2023), and others (Gidaris et al., 2018). The release of certain model weights, like CLIP (Cherti et al., 2023; Radford et al., 2021), ImageBind (Girdhar et al., 2023), and DeBERTa (He et al., 2020), has inspired us to employ them in our area of interest, discriminative multi-modal learning. Indeed, their integration has significantly enhanced the performance of various methods across different datasets.

**Discriminative Multi-Modal Learning.** Multi-modal learning has been proven to be superior to uni-modal learning in various areas (Zhang et al., 2023b; Xiao et al., 2020; Huang et al., 2021). However, challenges like Modality Competition (Huang et al., 2022b) or Modality Laziness (Du et al., 2023) often hinder the effectiveness of multi-modal joint training in sufficiently capturing uni-modal features (Peng et al., 2022; Fan et al., 2023; Wu et al., 2022). In practice, this has led to instances where multi-modal models empirically perform worse (Wang et al., 2020), even when uni-modal models employ the same encoder as the multi-modal counterparts. While some methods try to address this issue with extra loss terms (Fan et al., 2023) or gradient control (Peng et al., 2022), encoders from multi-modal learning still underperform compared to uni-modal training. Du et al. (2023) introduces Uni-Modal Ensemble (UME) to average uni-modal predictions, but it lacks cross-modal adaptation, limiting its performance.

**Parameter-Efficient Fine-Tuning (PEFT).** Parameter-Efficient Fine-Tuning is a methodology designed to fine-tune only a subset of parameters in large language models, enhancing their adapt-

Table 2: **Selection of Pre-trained Encoder**. (1) All visual encoders are from OpenCLIP(Cherti et al., 2023), where ViT-B is pre-trained on LAION-2B, and ViT-L is pre-trained on DataComp-1B. (2) All audio encoders are from Imagebind(Girdhar et al., 2023). (3) The text encoders BERT and DeBERTa come from Devlin et al. (2018) and He et al. (2020) respectively. (4) The encoder for optical flow is a ResNet-18 (He et al., 2016) pre-trained on ImageNet.

| Dataset
Encoder | AVE | KS | CREMA-D | Food101 | MM-IMDB | UCF101 |
|---|---|---|---|---|---|---|
| Visual | ViT-B | ViT-B | ViT-B | ViT-L | ViT-L | ViT-B |
| Audio/Text/Flow | ViT-B | ViT-B | ViT-B | BERT-B | DeBERTa-L | ResNet-18 |

Table 3: **Top-1 test accuracy (in %) of linear evaluation on encoders from multi-modal training** and uni-modal fine-tuned models on AVE, Kinetics-Sound and CREMA-D. The one with better performance is highlighted in **bold**.

| Encoder Source | AVE | | Kinetics-Sound | | CREMA-D | |
|---|---|---|---|---|---|---|
| | RGB | Audio | RGB | Audio | RGB | Audio |
| Multi-Modal Training | 83.3 | 79.8 | 78.7 | 67.9 | 70.3 | 64.8 |
| Uni-Modal Fine-Tuned | **88.1** | **85.6** | **84.3** | **69.6** | **77.7** | **75.8** |

ability to downstream tasks (Houlsby et al., 2019; Hu et al., 2021; Zhang et al., 2023a; Liu et al., 2023b; 2021; Dettmers et al., 2023). Among these methodologies, LoRA (Hu et al., 2021) is the most notable, which introduces novel trainable rank decomposition matrices to facilitate adaptation to new tasks. Once training is completed, these parameters can be seamlessly merged with the existing ones without incurring any additional inference cost. Previous works have also adopted similar approaches, using a linear projection layer (Liu et al., 2023a) or a querying transformer (Li et al., 2023a; Dai et al., 2023) to link a large visual model with a large language model. However, these studies primarily focuses on textual dialogues or generation, while our paper mainly focus on discriminative multi-modal learning.

## 3 ANALYSIS AND METHOD

In this section, we first illustrate that even when employing large-scale pre-trained models for multi-modal joint training, they can still suffer from insufficient learning of uni-modal features. We then introduce our proposed method, **M**ulti-**M**odal **Lo**w-**R**ank **A**daptation learning (**MMLoRA**), to address this issue.

### 3.1 MULTI-MODAL LEARNING WITH LARGE-SCALE PRE-TRAINED MODELS

Large-scale pre-trained models have demonstrated impressive performance in downstream tasks (Radford et al., 2021). With an increasing number of these models being open-sourced (Cherti et al., 2023; Girdhar et al., 2023; He et al., 2020), it motivates us to investigate how to further enhance discriminative multi-modal learning with them.

Table 2 displays the encoders chosen for various datasets. For a more comprehensive discussion and additional details, please refer to Sec 4.1.2. It's important to note that for specific downstream tasks, an additional linear layer is necessary to map the extracted features to the label space.

**Finetuned pre-trained models with uni-modal data outperform previous multi-modal models.** We first directly fully fine-tune these pre-trained models using the uni-modal data, with results presented in Table 1. In these four datasets (AVE, Kinetics-Sound, CREMA-D and MM-IMDB), fine-tuning a uni-modal model already outperforms the multi-modal models from recent publications (Fan et al., 2023; Peng et al., 2022; Li et al., 2023b). This encourages us to further investigate how to better utilize these models in discriminative multi-modal learning.

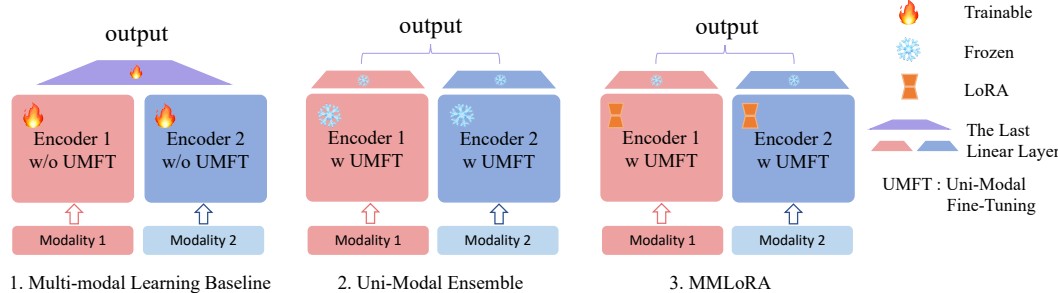

Figure 1: **Comparison of different multi-modal learning methods**. (1) The *baseline method* directly uses large-scale pre-trained encoders to extract features of the corresponding modality. Then, it concatenates the features from different modalities and passes them through a linear layer to obtain the final prediction; (2) *Uni-Modal Ensemble* (Du et al., 2023) first fine-tunes the respective models with uni-modal data separately, and then directly averages the outputs from different modalities to obtain the final prediction; (3) *MMLoRA* first freezes the uni-modal fine-tuned models and introduces extra trainable rank decomposition matrices to *a specific modality or all modalities' models*. It then performs multi-modal joint training, allowing the uni-modal fine-tuned models to better adapt across modalities, further enhancing the overall performance.

**Insufficient learning of uni-modal features in multi-modal learning.** In multi-modal joint training, we use pre-trained models to encode their respective modalities, obtaining their features. We then concatenate the features from different modalities and pass them through a linear layer to produce a prediction. Through end-to-end training, we obtain a multi-modal model. The process is illustrated in the leftmost subfigure of Figure 1. This approach is also referred to as late-fusion learning baseline and is widely used in multi-modal learning (Du et al., 2023). Its advantage is its applicability to a wide variety of different encoders. Then we employ linear evaluation (Chen et al., 2020b), which trains a linear layer on frozen encoders, to assess their feature extraction ability. As Table 3 shows, all encoders from multi-modal training are worse than their uni-modal fine-tuned counterparts. *Multi-modal learning with large-scale pre-trained encoders still suffer from insufficient learning of uni-modal features.* This result is consistent with the scenario when using backbones that are either not pre-trained or pre-trained on ImageNet (Peng et al., 2022), which has been proven to impact the model's generalization ability (Du et al., 2023).

In the following subsection, we introduce our proposed method to address this issue.

## 3.2 MULTI-MODAL LOW-RANK ADAPTATION LEARNING (MMLoRA)

Multi-modal joint learning struggles to fully capture uni-modal features, and there have been previous papers attempting to address this issue (Du et al., 2023; Fan et al., 2023; Peng et al., 2022). However, whether by introducing new loss functions (Fan et al., 2023) or dynamically balancing the learning progress of different modalities (Peng et al., 2022), none can enable the encoders from the multi-modal training to achieve performance equivalent to that from uni-modal training. Du et al. (2023) points out that directly training uni-modal models separately and then averaging the outputs of the uni-modal models is already quite strong (Uni-Modal Ensemble, as shown in Figure 1). However, (Du et al., 2023) also points out that UME is not suitable for all scenarios, as it cannot learn the so-called *paired features*. As can be seen from Section 3.2 of the paper (Du et al., 2023), in multi-modal datasets, different modalities need to adapt to each other to produce better results.

**MMLoRA.** Uni-Modal Fine-tuned models need further adaptation to each other to achieve better results. Given that uni-modal fine-tuned models have already learned a significant amount of features, our hypothesis is that only a small portion of parameters is needed for this adaptation. To this end, we draw inspiration from Parameter-Efficient Fine-Tuning (PEFT), particularly from LoRA (Hu et al., 2021) and propose **M**ulti-**M**odal **Lo**w-**R**ank **A**daptation learning (**MMLoRA**). Specifically, we freeze the weights of the uni-modal fine-tuned models and introduce additional trainable rank decomposition matrices to a specific modality or all modalities' models. We then utilize multi-modal

joint training to train these new parameters, enabling the modalities to better adapt to each other. Specifically, we average the uni-modal predictions to obtain a multi-modal prediction, then compute the loss against the label, from which we derive the gradients to update the LoRA parameters. We also illustrate MMLoRA in Figure 1.

**Formal Definition of MMLoRA.** We suppose there are $M$ modalities within the problem. For each single modality $m = 1, ..., M$, we have a dataset $\mathcal{Z}^m = \left\{ \left( x_j^m, y_j \right) \right\}_{j=1,...,N}$ containing the training data pairs of input $(x^m)$ and label $(y)$. The datasets with all modalities are denoted as $\bar{\mathcal{Z}}$. Then we denote the uni-modal model for each single modality with its weight $\Phi^m$ as $P_{\Phi^m}(y|x^m)$, which will be short as $P_\Phi^m(y|x)$ for simplicity. We first apply Uni-Modal Fine-Tuning (UMFT) by optimizing each encoder with its own uni-modal data separately with fully fine-tuning, which updates the whole model by repeatedly following the gradient of

$$\max_{\Phi^m} \sum_{(x,y) \in \mathcal{Z}^m} \log \left( P_\Phi^m(y|x) \right), m = 1, ..., M. \tag{1}$$

Recalling the reparameterization scheme of LoRA (Hu et al., 2021), for a pre-trained weight matrix $W_0 \in \mathbb{R}^{d \times k}$, the parameter adaption $\Delta W$ can be constrained in a low-rank decomposition as

$$W_0 + \Delta W = W_0 + BA, \tag{2}$$

where $A \in \mathbb{R}^{r \times k}$ and $B \in \mathbb{R}^{d \times r}$ are matrices with rank $r \ll \min(d, k)$. $A$ is Gaussian initialized and $B$ is set as azero matrix to ensure the update $\Delta W$ is zero at the beginning of training.

In MMLoRA, we apply the LoRA reparameterization scheme (2) to uni-modal fine-tuned models $\Phi_{\text{UMFT}}^m$ obtained by Eq. 1. For each uni-modal fine-tuned model reparametrized by LoRA, the real trainable parameters $\Theta^m$ are significantly fewer than the total parameters in $\Phi^m$, inheriting the advantage of LoRA in parameter efficiency and preventing overfitting. Once training is completed, these newly introduced parameters can be directly merged with the original parameters (as Eq. 2 shows). We denote the LoRA update of each uni-modal model as $\Delta \Phi^m = \Delta \Phi(\Theta^m)$. Empirically, we have found that simply allowing only one modality to be reparameterized by LoRA, and then updating this modality's LoRA parameters to adapt another modality through multi-modal joint learning, is also very effective. To this end, we have the flexibility to selectively update specific $\Theta^m$. We denote the collection of selected indexes $m$ of each modality as the set $\mathcal{M}$. The selected parameters $\{\Theta^m\}_{m \in \mathcal{M}}$ to be updated are collectively represented as $\Theta$. Then we optimize the multi-modal joint training objective (3) towards $\Theta$ with the entire multi-modal dataset $\bar{\mathcal{Z}}$ as

$$\max_\Theta \sum_{(x,y) \in \bar{\mathcal{Z}}} \log \sum_{m=1}^M P_{\Phi_{\text{UMFT}} + \Delta \Phi(\Theta)}^m (y|x). \tag{3}$$

Finally, we obtain the MMLoRA model as $P^{\text{MMLoRA}}(y|x) = \frac{1}{M} \sum_{m=1}^M P_{\Phi_{\text{UMFT}} + \Delta \Phi(\Theta)}^m (y|x)$, where $P_{\Phi_{\text{UMFT}} + \Delta \Phi(\Theta)}^m$ denotes a uni-modal model firstly trained by Uni-Modal Fine-Tuning (UMFT, 1), then reparameterized through LoRA, followed by a multi-modal joint training (3).

Overall, MMLoRA introduces a limited number of new parameters for the uni-modal fine-tuned models, which provides an opportunity for different modalities to adapt to each other more effectively, ultimately leading to improved performance.

In the following section, we will demonstrate the effectiveness of MMLoRA and conduct ablation experiments to gain a deeper understanding of the underlying principles that drive its success.

## 4 EXPERIMENT

In this section, we begin by describing the six multi-modal datasets (including AVE, Kinetics-Sound, CREMA-D, MM-IMDB, UPMC Food101 and UCF101) we use, pre-trained models we select, and other settings. Subsequently, we present our primary experimental results, demonstrating the significant performance improvement by using pre-trained large models and the effectiveness of MMLoRA across a range of multi-modal datasets. Lastly, we conduct ablation studies on MMLoRA to further understand the underlying mechanisms of its operation.

## 4.1 Datasets and Experimental Settings

### 4.1.1 Datasets

**1. Audio-Visual Datasets:** (1) The *Audio-Visual Event localization (AVE) dataset*, as introduced in Tian et al. (2018), contains 4,143 unrestricted 10-second videos across 28 event types such as Rodents, Accordion, Mandolin and so on. AVE is a subset of AudioSet (Gemmeke et al., 2017). The train/val/test splits are 3,339/402/402 videos, respectively; (2) The *Kinetics-Sounds* dataset, referenced in (Arandjelovic & Zisserman, 2017), is a curated subset of Kinetics400, which features YouTube videos with hand-labeled human actions. This subset encompasses 32 classes, including actions like playing the harmonica, tapping a pen, shoveling snow and so on. The dataset is divided into 22,728 training videos and 1,593 validation videos. (3) The *CREMA-D* dataset (Cao et al., 2014), is an audio-visual dataset designed for speech emotion recognition. It features 7,442 short video clips (2-3 seconds each) from 91 actors expressing six common emotions: angry, happy, sad, neutral, discarding, disgust, and fear. Emotion labels were sourced from 2,443 crowd-sourced raters. The dataset is split into a 6,698-sample training set and a validation set in a 9:1 ratio, with a separate 744-sample test set.

**2. Vision-Language Datasets:** (1) The *MM-IMDB* dataset, introduced by Arevalo et al. (2017), combines movie plot outlines with movie posters for genre classification. Each movie may belong to multiple genres, making it a multilabel prediction task. The dataset was created to address the limited availability of quality multi-modal classification datasets. The train/val/test splits are 15,552/2,608/7,799 videos, respectively; (2) The UPMC FOOD101 dataset, introduced by Wang et al. (2015), offers textual descriptions of recipes spanning 101 food categories. These descriptions, extracted from curated web pages, are paired with an image sourced from Google Image Search, which might occasionally match a noisy category. The goal is to determine the appropriate food label for each text-image pair. The train/test splits are 67,972/22,716 videos, respectively.

**3. RGB-Optical Flow Dataset, UCF101.** The UCF101 dataset, presented by Soomro et al. (2012), is designed for action recognition, featuring 101 distinct action categories. It contains around 7k training videos and 3k testing videos. For our experiments, we employ the RGB and flow data supplied by Feichtenhofer et al. (2016).

### 4.1.2 Experimental Settings

**Selection of Pre-Trained Encoders.** In Table 2, we display the encoders we selected for different datasets. On the audio-visual dataset, we employ ViT-B as the visual encoder, as it is already quite effective, outperforming previous multi-modal models. Although CLIP has trained a text encoder aligned with images, its text encoder is not as effective as those directly pre-trained on text (Saharia et al., 2022). Therefore, we choose for text encoders that were purely pre-trained on text, such as BERT and DeBERTa. For UPMC Food101, we select BERT-Base as various text encoders show similar results. To better demonstrate the effectiveness of MMLoRA in comparison to the baseline methods, we also implement MMLoRA using the same encoders as the baselines for a direct comparison. In such cases, we will point it out. Additionally, we discover that bigger models are not necessarily better. We've included more details in the Appendix B.

**Data Preprocessing.** (1) For *images*, we randomly resize to 224x224, then apply horizontal flip, and normalize them using the OpenCLIP MEAN and STD values. In audio-visual datasets, we randomly take 3 frames as input when training, and put them into the 2D network as Peng et al. (2022) does; (2) For *audio*, we first ensure a sample rate of 16k, then convert the t-second audio waveform into 128-dimensional log Mel filterbank sequences. Our audio preprocessing is consistent with that of Girdhar et al. (2023) or Gong et al. (2021); (3) For *text*, we use the corresponding tokenizer for different language models. Additionally, we set the maximum token sequence length to 512.

**Optimizer.** When fine-tuning the language model in a uni-modal setting, we use BertAdam (Devlin et al., 2018) as the optimizer; We use SGD when training the optical flow in UCF101; For all other experiments, we employ AdamW (Loshchilov & Hutter, 2017). For the Vision-language task, we use a batch size of 160, while for other tasks, we use a batch size of 64.

**MMLoRA Settings.** Unless oterwise specified, we set the rank of LoRA to 1. For transformer (Vaswani et al., 2017) models, we add LoRA to all Query, Key, and Value layers. For ResNet (He et al., 2016), we apply LoRA to all convolutional layers. For the CREMA-D

Table 4: **Comparison of linear evaluation and fully fine-tuning** of pre-trained models on uni-modal data of AVE, Kinetics-Sound (KS), CREMA-D, MM-IMDB and UPMC Food101. The reported evaluation metric are Top-1 Accuracy (AVE, Kinetics-Sound , CREMA-D and UPMC Food101) and F1-Micro (MM-IMDB). Better performance is highlighted in **bold**.

| Method | AVE | | KS | | CREMA-D | | MM-IMDB | | Food101 | |
|---|---|---|---|---|---|---|---|---|---|---|
| | RGB | Audio | RGB | Audio | RGB | Audio | RGB | Text | RGB | Text |
| linear eval | 87.6 | 82.1 | 79.0 | 66.2 | 51.2 | 60.6 | 48.7 | 22.1 | 81.8 | 21.2 |
| fine-tuning | **88.1** | **85.6** | **84.3** | **69.6** | **77.7** | **75.8** | **60.2** | **68.6** | **84.3** | **86.6** |

Table 5: **Top-1 Test Accuracy of different methods on Audio-Visual Datasets** (AVE, Kinetics-Sound and CREMA-D). *Avg Acc* represents the average accuracy across the three datasets. * indicates our implementation. The best performance under same backbones is highlighted in **bold**.

| Method | Backbone (A/V) | AVE | KS | CREMA-D | Avg. Acc. |
|---|---|---|---|---|---|
| G-Blending (Wang et al., 2020) | ResNet18/ResNet18 | 65.5 | 62.2 | 58.7 | 62.1 |
| OGM-GE (Peng et al., 2022) | ResNet18/ResNet18 | 76.9 | 63.1 | 62.2 | 67.4 |
| PMR (Fan et al., 2023) | ResNet18/ResNet18 | 74.3 | - | 65.3 | - |
| UME* (Du et al., 2023) | ResNet18/ResNet18 | 85.4 | 78.8 | 78.2 | 80.8 |
| **MMLoRA** (ours) | ResNet18/ResNet18 | **86.9** | **79.4** | **81.9** | **82.7** |
| Multi-Modal Baseline* | ViT-B/ViT-B | 94.7 | 90.6 | 87.6 | 91.0 |
| Classifier on frozen features* | ViT-B/ViT-B | 93.7 | 90.1 | 85.3 | 89.7 |
| MBT (Nagrani et al., 2021) | ViT-B/ViT-B | - | 85.0 | - | - |
| OGM-GE* (Peng et al., 2022) | ViT-B/ViT-B | 95.5 | 90.4 | 88.4 | 91.4 |
| UME* (Du et al., 2023) | ViT-B/ViT-B | 95.4 | 90.8 | 87.8 | 91.3 |
| Fully Fine-tuned UME* | ViT-B/ViT-B | 95.2 | 91.3 | 87.5 | 91.3 |
| **MMLoRA** (ours) | ViT-B/ViT-B | **96.2** | **91.4** | **88.6** | **92.1** |

dataset, we only reparametrize the audio model using LoRA. For UCF101, only the optical flow model is reparametrized with LoRA. In other cases, the results we report for MMLoRA involve reparametrization using LoRA for all modalities. We further conduct an ablation study on this aspect in Sec 4.3.

**Other Hyper-parameters.** In this sub-section, we've outlined some common settings. There are other settings, such as the learning rate, that vary by task. Due to space constraints, we have placed other hyper-parameters in Appendix A.

## 4.2 MAIN EXPERIMENTAL RESULTS

### 4.2.1 FOR LARGE-SCALE PRE-TRAINED MODELS: DIRECTLY USE THEIR FEATURES OR PERFORM FINE-TUNING?

Large-scale pre-trained models have proven to exhibit strong zero-shot performance (Radford et al., 2021). However, it seems that only when both the data volume and model size reach a considerably large scale does zero-shot outperform fine-tuning (OpenAI, 2023). In this sub-section, we first compare two methods using our data: the first approach involves adding a linear layer to the pre-trained models and training only this layer; the second approach entails fully fine-tuning the models. Note that for this experiment, we train each model only on uni-modal data. The results are shown in the Table 4. Fine-tuning pre-trained models on uni-modal data significantly outperforms linear evaluation. Thus, in our subsequent experiments, we won't directly use features extracted from the pre-trained model; instead, we will train the encoeder's parameters with our data.

### 4.2.2 THE EFFECTIVENESS OF LARGE-SCALE PRE-TRAINED MODELS

We present the results of different methods using various backbones on different datasets in Tables 5, 6, and 7. In these three tables, the results below the dashed line use stronger large-scale pre-trained

Table 6: **The performance of MMLoRA and other methods on UPMC Food101 (Acc.) and MM-IMDB (F1-Micro/F1-Macro)**. Above the dashed line, the backbone used is consistent with Kiela et al. (2019), while below the dashed line, the methods compared are reimplemented using improved backbone by Li et al. (2023b).

| Method | Food101 | MM-IMDB |
|---|---|---|
| MMBT | 92.1 | 66.8/61.6 |
| **MMLoRA** (ours) | **93.2** | **67.2/61.7** |
| Baseline | 93.29 | 64.9/59.6 |
| PMF | 91.51 | 64.5/58.8 |
| PMF-L | 91.68 | 66.7/61.7 |
| MBT | 93.6 | 64.8/59.6 |
| MMBT | 94.10 | 66.1/60.8 |
| **MMLoRA** (ours) | **95.9** | **71.7/67.5** |

Table 7: **Top-1 Test Accuracy (in %) of different methods on UCF101.** * indicates our implementation and the performance of the other methods is derived from Du et al. (2023). We also present the backbones used for different methods (RGB/Optical Flow).

| Method | Backbone | Acc. |
|---|---|---|
| MM Baseline | Res18/Res18 | 82.3 |
| G-Blending | Res18/Res18 | 83.0 |
| OGM-GE | Res18/Res18 | 84.0 |
| UME | Res18/Res18 | 86.8 |
| **MMLoRA** | Res18/Res18 | **87.1** |
| UME* | ViT-B/Res18 | 93.0 |
| **MMLoRA** | ViT-B/Res18 | **93.4** |

Table 8: **Top-1 Test Accuracy (in %) of linear evaluation on encoders from MMLoRA** and uni-modal fine-tuned models on AVE, Kinetics-Sound and CREMA-D. In this experiment, the encoders from MMLoRA are all re-parameterized with $r = 1$ and then performed multi-modal joint training.

| Encoder Source | AVE | | Kinetics-Sound | | CREMA-D | |
|---|---|---|---|---|---|---|
| | RGB | Audio | RGB | Audio | RGB | Audio |
| Uni-Modal Fine-Tuned | 88.1 | 85.6 | 84.3 | 69.6 | 77.7 | **75.8** |
| **MMLoRA** (ours) | 88.1 | **86.4** | **85.3** | **69.8** | **78.1** | 75.7 |

encoders than those above. Notably, especially in Table 5 and 7, the performance of the methods below the dashed line far surpasses that of the methods above. In Table 6, with improved backbones, MMLoRA's performance is also significantly enhanced. The evident performance gap highlights the necessity of using large-scale pre-trained models.

### 4.2.3 MAIN RESULTS OF MMLoRA

In this section, we demonstrate the effectiveness of MMLoRA across three different types of datasets and various backbones.

**MMLoRA on audio-visual datasets.** As Table 5 shown, we can observe that MMLoRA consistently exhibits the best performance across different backbones. The methods compared include balancing training by adding loss (G-Blending (Wang et al., 2020), PMR (Fan et al., 2023)), adjusting the training progress of different modalities by modifying gradients (OGM-GE (Peng et al., 2022)), novel fusion framework (MBT (Nagrani et al., 2021)),or directly averaging uni-modal predictions (UME (Du et al., 2023)). We also compare multi-modal joint full fine-tuning of the uni-modal fine-tuned models (namely *Fully Fine-tuned UME*), and the results are inferior to MMLoRA. We hypothesize that the limited data, combined with the excessive model parameters, makes full fine-tuning less effective than parameter-efficient fine-tuning. *Classifier on frozen features* refers to using uni-modal fine-tuned models to extract features and training a multi-modal linear layer. And the multi-modal baseline method is illustrated in the leftmost subfigure of Figure 1.

**MMLoRA on vision-language datasets.** Here, we compare MMLoRA with MBT (Nagrani et al., 2021), MMBT (Kiela et al., 2019), and PMF (Li et al., 2023b). When we use the same backbone as the original MMBT paper, MMLoRA outperforms MMBT. Additionally, when we employ a superior backbone, MMLoRA also surpasses various methods reimplemented by Li et al. (2023b).

**MMLoRA on UCF101.** The Uni-Modal Ensemble (Du et al., 2023) has already proven to be very effective on UCF101. As Table 7 shows, we apply MMLoRA to UCF101 to further boost performance, regardless of the backbone used.

Table 9: **Comparison between the MMLoRA model where LoRA is applied to a single modality and the MMLoRA model where LoRA is applied to all modalities** on AVE, Kinetics-Sound and CREMA-D. *RGB* or *Audio* indicates that we only re-parameterize the fine-tuned RGB or Audio Model by LoRA, and then, through multi-modal joint training, we allow the specified modality to adapt to the other. *Both* means that we apply LoRA re-parameterization to both modalities and then jointly train them. The one with better performance is highlighted in **bold**.

| Method | AVE | | | Kinetics-Sound | | | CREMA-D | | |
|---|---|---|---|---|---|---|---|---|---|
| | RGB | Audio | Both | RGB | Audio | Both | RGB | Audio | Both |
| **MMLoRA** | 95.7 | **96.2** | **96.2** | 91.3 | 91.3 | **91.4** | 87.9 | **88.6** | 88.3 |

Table 10: **Top-1 Test Accuracy (in %) of MMLoRA with different ranks** on the Kinetics-Sound and AVE datasets. We include the results of UME here for comparison.

| Rank ($r$)
Dataset | UME | 1 | 2 | 4 | 8 | 64 |
|---|---|---|---|---|---|---|
| AVE | 95.4 | 96.2 | **96.5** | **96.5** | 96.2 | 95.2 |
| Kinetics-Sound | 90.8 | **91.4** | 91.3 | **91.4** | 91.3 | 91.1 |

### 4.3 ABLATION STUDY ON MMLoRA

**MMLoRA does not affect the uni-modal feature extraction.** MMLoRA introduces new trainable parameters to the uni-modal fine-tuned models and conducts multi-modal joint training. Taking the encoders trained by MMLoRA for linear evaluation, as shown in Table 8, not only does MMLoRA not affect the extraction of uni-modal features, but in some cases, it even surpasses its uni-modal training counterpart.

**Apply LoRA re-parameterizion to which part?** In Table 9, we experiment with adding LoRA to just one uni-modal fine-tuned model (during this process, we essentially adjust only the LoRA parameters of a particular modality to make it adapt to another modality.) and compare it to that of applying LoRA to both models. Firstly, we can see that all of the resulting models always outperform Uni-Modal Ensemble (Refer to the Table 5). On the CREMA-D dataset, applying LoRA solely to the Audio Model, followed by multi-modal joint training, yields better results. And for the other two datasets, directly applying LoRA to both modalities is superior. We hypothesize that allowing only one modality to update its parameters to adapt to the other modality might lead to more stable training in some scenarios.

**Rank Selection in MMLoRA.** We also conduct an ablation study on the rank values of MMLoRA on the Kinetics-Sound and AVE datasets. As shown in Table 10, when the rank is set too high, the performance declines (in AVE, the performance of MMLoRA with a rank of 64 might even be inferior to UME). This might be due to the introduction of too many parameters while the dataset isn't large enough. Although setting the rank to 1 doesn't always yield the best results, the performance is consistently good. These results suggests that cross-modal adaptation is necessary and might not require many parameters, and thus, in our main experiments with MMLoRA, we set the rank to 1.

**Other Ablation Study.** We try reparametrizing pre-trained models with LoRA directly and then conduct multi-modal joint training. Results are in the Appendix C due to space constraints.

## 5 CONCLUSION

Large-scale pre-trained models have undoubtedly significantly enhanced discriminative multi-modal learning, yet they also encounter the issue of sufficient learning of uni-modal features. We introduce **M**ulti-**M**odal **Lo**w-**R**ank **A**daptation learning (**MMLoRA**), which adds a small amount of parameters to uni-modal fine-tuned models and then engages in multi-modal joint training to better adapt across modalities, thereby boosting the overall performance. We hope this paper can bring some new insights to the field of discriminative multi-modal learning.

## REPRODUCIBILITY STATEMENT

In our paper, we outline the data, model, and training hyperparameters used, as detailed in Sec 4.1.1 and Appendix A. Since our approach is straightforward (Sec 3.2), it ensures a high level of reproducibility for our work. Our code is also available in the supplementary material.

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

# A  ADDITIONAL EXPERIMENTAL SETTINGS

## A.1  EXPERIMENTAL SETTINGS ON AUDIO VISUAL DATASETS

**Settings of Uni-Modal Training.**  For uni-modal fine-tuning with large-scale pre-trained models, we add a linear layer for feature-to-label mapping and use a learning rate of $1e-5$. For training with ResNet18, the learning rate is set to $1e-4$.

**Settings of Multi-Modal Training.**  For multi-modal joint training, we use a learning rate of $1e-5$. During MMLoRA's adaptation phase, the rate is $1e-4$ with large uni-modal fine-tuned models, and $1e-6$ with ResNet18 backbones.

## A.2  EXPERIMENTAL SETTINGS ON VISION LANGUAGE DATASETS

**Common Settings.**  We reduce the learning rate when a metric has stopped improving. If the validation accuracy does not increase for two consecutive epochs, we reduce the learning rate to half of its original value.

**Settings of Uni-Modal Training.**  For fine-tuning language models and ResNet, the learning rate is 5e-5. For ViT-L models, it's 1e-5. Noting that for vision-language datasets, ResNet152 is our baseline visual backbone.

**Settings of Multi-Modal Training.**  For MMLoRA's multi-modal adaptation with large-scale uni-modal fine-tuned models, we use a learning rate of $5e-4$ on UPMC Food101 and $5e-6$ on MM-IMDB. With ResNet152 and BERT-base, it's $1e-4$ on UPMC Food101 and $1e-5$ on MM-IMDB.

## A.3  EXPERIMENTAL SETTINGS ON ACTION RECOGNITION DATASETS (UCF101)

When training ViT-B on RGB data, we set the learning rate to $1e-5$. For MMLoRA's multi-modal adaptation with ViT-B (RGB) and ResNet18 (optical flow), the rate is $1e-4$. With ResNet18 for both RGB and optical flow, it's $1e-5$. Other settings align with Du et al. (2023).

# B  IS BIGGER ALWAYS BETTER FOR MODELS?

When selecting more suitable pre-trained models for different datasets, we find that in some cases, bigger models do not necessarily yield better results. For instance, on the UPMC Food101 dataset, we select four different language models that vary both in pre-trained data and model size. However, their performance seems to be comparable as shown in Table 11. Thus, for the text encoder of this dataset, we used BERT-Base. In CREMA-D, the smaller ViT-Base outperforms ViT-Large, as shown in Table 12. At the same time, ViT-B performs sufficiently well on other audio-visual datasets, even surpassing previous multi-modal methods. Therefore, we choose ViT-Base for the Audio-Visual datasets.

Table 11: **Top-1 Test Accuracy of different language models fine-tuned on text modality of UPMC food101.**

| Model 
 Dataset | BERT-Base | BERT-Large | DeBERTa-Base | DeBERTa-Large |
|---|---|---|---|---|
| UPMC food101 | 86.6 | 86.7 | 86.6 | 86.8 |

# C  IMPLEMENT MMLoRA DIRECTLY ON PRE-TRAINED MODELS

In this section, we conduct another ablation study on MMLoRA: directly reparametrizing the pre-trained models using LoRA and then proceeding with multi-modal joint training. Compared to the

Table 12: **Top-1 Test Accuracy of different visual models fine-tuned with visual data of CREMA-D.**

| Model
Dataset | ViT-B | ViT-L |
|---|---|---|
| CREMA-D | 77.7 | 75.5 |

Table 13: **Top-1 Test Accuracy of MMLoRA (with UMFT and without UMFT) on AVE, Kinetics-Sound and CREMA-D**. We set rank of MMLoRA as 1 and UMFT represents Uni-Modal Fine-Tuning. Note that in this table, MMLoRA refers to reparametrizing both modality models using LoRA.

| Dataset
Method | AVE | KS | CREMA-D |
|---|---|---|---|
| MMLoRA (w/o UMFT) | 94.0 | 89.5 | 83.7 |
| MMLoRA (w UMFT) | **96.2** | **91.4** | **88.3** |

MMLoRA implementation in Sec 4, this skips the uni-modal fine-tuning for the pre-trained models. As shown in Table 13, MMLoRA with uni-modal fine-tuning performs much better than without it, which highlights the importance of uni-modal fine-tuning.

