# OpenReview forum: "Improving Discriminative Multi-Modal Learning with Large-Scale Pre-Trained Models"
_ICLR.cc/2024/Conference — ICLR 2024 Conference Withdrawn Submission_

### Official Review · Reviewer_32xM · 2023-10-31

**Soundness:** 2 fair
**Presentation:** 2 fair
**Contribution:** 2 fair
**Rating:** 3
**Confidence:** 5

**Summary:**

When applying large-scale pre-training models to multi-modal joint training, it can lead to insufficient feature learning of unimodal, and even perform worse than the performance of unimodal training alone, thereby weakening the generalization ability of multi-modal models. Therefore, the proposed method first freezes the weights of the unimodal fine-tuning model and introduces additional trainable rank decomposition matrices (LORA) into the model of a specific modality or all modalities. Then, these new parameters are trained through multi-modal joint training, allowing various modalities to better adapt to each other.

**Strengths:**

The paper proposes a method called Multi-Modal Low-Rank Adaptation learning (MMLoRA), which introduces trainable low-rank decomposition matrices in multi-modal training, allowing for better adaptation between different modalities, thereby improving the performance of multi-modal learning.
The effectiveness of MMLoRA has been demonstrated on multiple datasets, including audio-visual datasets (AVE, Kinetics-Sound, CREMA-D), visual-language datasets (MM-IMDB, UPMC Food101), and RGB-optical flow action recognition datasets (UCF101).

**Weaknesses:**

The paper lacks innovation and novelty.
The biggest shortcoming of the paper is that it does not explain why the proposed MMLoRA method can address the problem of insufficient feature learning of unimodal under the condition of multi-modal joint training.
In addition, from the experimental results, the performance improvements are limited, which is not enough to prove that the LoRA fine-tuning method can address this problem.
The reason for the effectiveness of the proposed method remains to be considered. Is it because LoRA's efficient fine-tuning method works, or because LoRA really solves the problem of insufficient learning of unimodal features? This paper cannot draw a conclusion and is unreliable.

**Questions:**

The paper lacks innovation and novelty.
The biggest shortcoming of the paper is that it does not explain why the proposed MMLoRA method can address the problem of insufficient feature learning of unimodal under the condition of multi-modal joint training.
In addition, from the experimental results, the performance improvements are limited, which is not enough to prove that the LoRA fine-tuning method can address this problem.
The reason for the effectiveness of the proposed method remains to be considered. Is it because LoRA's efficient fine-tuning method works, or because LoRA really solves the problem of insufficient learning of unimodal features? This paper cannot draw a conclusion and is unreliable.

---

### Official Review · Reviewer_buaj · 2023-11-01

**Soundness:** 1 poor
**Presentation:** 1 poor
**Contribution:** 1 poor
**Rating:** 3
**Confidence:** 4

**Summary:**

This paper presents a method for improving multi-modal learning by leveraging large-scale pre-trained uni-modal models. The proposed Multi-Modal Low-Rank Adaptation learning (MMLoRA) freezes the weights of uni-modal models, adds extra trainable rank decomposition matrices, and then carries out multi-modal joint training, to enhance adaptation between modalities, thereby improving overall performance. The effectiveness of MMLoRA is demonstrated across three dataset categories: audio-visual, vision-language, and RGB-Optical flow.

**Strengths:**

- The paper shows a slight performance improvement over the uni-modal ensemble (UME) method, demonstrating the efficacy of the proposed Multi-Modal Low-Rank Adaptation learning (MMLoRA).

- The study is innovative in using additional fine-tuning of UME, which actually resulted in enhanced performance.

**Weaknesses:**

- The use of an adapter with a small number of parameters to conduct fine-tuning when there is insufficient data in the target task is a common method. Unfortunately, this paper also uses fine-tuning with the LoRA adapter but doesn't offer a special design or consideration for multi-modal situations.

- The performance of the proposed method appears to be highly dependent on LoRA's rank after close examination of the experimental results. Despite the utmost importance of the relationship between LoRA rank or full fine-tuning and data size, there is lack of discussion or study on this.

- Unfortunately, this paper reads like a technical report which simply applies LoRA to the uni-modal ensemble method and checks the performance change, thus missing depth and more extensive analysis.

**Questions:**

- Why should LoRA be used? What effects do other parameter-efficient fine-tuning (PEFT) methods have?

- Are there any unique challenges or features to be considered when applying PEFT methods like LoRA in multi-modal problems? The paper could provide more detailed explanations or possible directions for future study on these topics.

---

### Official Review · Reviewer_ChZV · 2023-11-02

**Soundness:** 3 good
**Presentation:** 3 good
**Contribution:** 2 fair
**Rating:** 6
**Confidence:** 3

**Summary:**

This paper investigates how to better leverage large-scale pre-trained uni-modal models to further enhance multi-modal learning. Then, a Multi-Modal Low-Rank Adaptation learning (MMLoRA) method is proposed to improve multi-modal learning. Experiments on three dataset categories  demonstrate the effectiveness of the proposed method.

**Strengths:**

Employing LoRA for multi-modal learning looks interesting.

The paper includes a few interesting analysis on different  uni-modal and multi-modal models.

The proposed MMLoRA method is effective.

**Weaknesses:**

The novelty is not that significant.

**Questions:**

No more questions.

---

### Official Review · Reviewer_iWgQ · 2023-11-07

**Soundness:** 3 good
**Presentation:** 3 good
**Contribution:** 3 good
**Rating:** 5
**Confidence:** 4

**Summary:**

The paper proposes Multi-Modal LowRank Adaptation learning (MMLoRA) to improve the multi-model performance with large pretrained models. The lightweight lora layers are introduced into the uni-model backbone to enhance the  adaption between modalities.  The audio, vision and language models are investigated to validate the performance of MMLoRA.

**Strengths:**

- The paper investigates MMLoRA with thorough experiments and ablation studies, such as the modality, the pretrained models and datasets. The effectiveness of large-pretrained models and lora layer is validated with performance improvement in multi-modal tasks.

**Weaknesses:**

- LoRA has been widely used in LLM and MLLM. The novelty of MMLoRA is a little limited as shown in Figure1 , and the experiment does not show impressive result in the multi-modal tasks.
- The paper shows the effectiveness of large-scale pretrained models with ResNet-18 and ViT-B. However, the size of model is relatively small for Multi-Modal model, and a much larger backbone should also be studied.
-  The introduction of lora in different parts of model can affect model performance as shown in section 4.3. The paper does not give reasonable explanations and enough experiments. How to apply lora in the multi-modal model for different tasks?

**Questions:**

Listed above

**Details Of Ethics Concerns:**

None.